# Target Velocity Ghost Imaging Using Slice Difference Method

**DOI:** 10.3390/s23094255

**Published:** 2023-04-25

**Authors:** Fan Jia, Zijing Zhang, Yuan Zhao

**Affiliations:** School of Physics, Harbin Institute of Technology, Harbin 150001, China; 19B911037@stu.hit.edu.cn

**Keywords:** ghost imaging, high resolution, slice difference

## Abstract

Ghost imaging is a technique that uses the correlation between reference and signal arms to obtain intensity images of targets. Compared with the existing laser active imaging methods, ghost imaging can improve the signal-to-noise ratio and resolution. In this paper, through the use of the slice difference method, we propose a new scheme that allows a velocity image of moving targets to be obtained. We conduct a complete theoretical analysis and provide a proof-of-principle experiment. The experimental results are in good agreement with those of the theoretical analysis, and a velocity image with 64 × 64 resolution is obtained. This protocol achieves a great increase in the signal-to-noise ratio over what would be achievable using direct imaging. The results show a fully functional instance of velocity imaging, which is a key advancement on the path towards the multi-dimensional information acquisition of moving targets. Our scheme fulfils an urgent need for the detection of moving targets and may thus find use in fields such as target attitude perception and security monitoring.

## 1. Introduction

Laser active detection has the advantages of a high resolution, compact size, strong anti-interference, and a wide application range. The detection of a target velocity image in laser active detection is a niche but valuable research field. At present, the methods of obtaining velocity information by laser active detection include a Doppler radar velocity measurement [1], a pulse time of flight velocity measurement, but there are some difficulties due to the limitations of volume and power consumption, and the sensitivity cannot meet the detection requirements. Traditional ghost imaging, which uses the spatial correlation of light to present intensity images of target [2,3,4], adopts a single point detection system. Its special imaging mechanism enables it to achieve high performance laser active imaging detection under the limitation of volume and power consumption [5,6,7,8]. At the same time, some recent studies have applied ghost imaging in communications [9,10].

Shaprio, of the MIT, pioneered computational ghost imaging and simplified the imaging system [11]. Han developed the first compressed sensing ghost imaging radar experiment based on sparse constraint [12] and verified the advantages of this scheme in imaging the range, rate, and spatial resolution. Subsequently, Yang Xu further optimized 3D correlation imaging and proposed the optimal slice number method (OSN) for the range profile [13,14]. At present, ghost imaging mainly focuses on the research of intensity images, and research has been carried out on the detection and performance improvement of some range images. The research on moving targets mainly focuses on eliminating the blur caused by motion [15,16]; therefore, in the follow-up research, the removal of the motion blur of a ghost imaging transverse target has become a research hotspot [17,18]. Yu, Yuan jin developed the imaging of transverse moving targets using artificial bee colony optimization [19]. Although there have been reports regarding the elimination of ghost imaging target motion blur, research on the velocity image of ghost imaging has not been reported. Velocity image detection has a high value in autopilot, space rendezvous and docking, and other application scenarios.

At present, there is no imaging system for velocity images; therefore, it is necessary to carry out ghost imaging velocity image research. In this paper, the detection method of a ghost imaging target velocity image based on slice difference is proposed. The results show that compared with the array velocity detection system, the system has a simple structure and stronger practical value, and it can greatly improve the imaging sensitivity. The ghost imaging velocity image provides a detection method for acquiring more dimensional image information of the target.

## 2. System Description and Method

On the basis of the traditional intensity image, we used laser pulse ranging to obtain the range image of the target. By setting a time slice, the pixel points of the range image at different times were differentiated, and the slice difference result was used for time integration to obtain the velocity image of the target.

The ghost imaging velocity image detection system based on slice difference is shown in Figure 1. The detection system selected a laser with a high laser intensity, strong directivity, and narrow pulse width. The single point detector received and detected the echo signal according to the time axis. We then divided the time axis evenly, sliced from $t_{0}$ time in a pulse time interval, and took the detected signal as the $t_{1}$ slice cut-off time until the $t_{n}$ slice cut-off time when the signal cannot be detected. We define the $t_{1}-t_{n}$ process as a detection cycle. The pulsed laser emitted by the laser can be expressed as:(1)$$P(t)=P_{0}\exp\left(-\frac{t^{2}}{2a^{2}}\right)$$
where $P_{0}$ is the peak power of the laser, and a is the pulse width of the laser pulse. The detector receives an echo signal focused as a single point. The function with target information is defined as $T\left(\vec{r}\right)$; the matrix distribution of the emission detection speckle light field output by the modulated pulse of the laser is $I\left(\vec{r}\right)$, the echo intensity reflected from the target received by the bucket detector is:(2)$$B(t)=I\left(\vec{r}\right)T\left(\vec{r}\right)P_{0}\exp\left(-\frac{t^{2}}{2a^{2}}\right)g\left(t\right)$$
where $g\left(t\right)=\sum_{n=1}^{N}\delta \left(t-t_{0}-nT\right)$ is the distance function, $t_{0}$ is the start time of the single point detector, and T is the time interval between the detection distance and the gate.

According to the ghost imaging intensity image formula:(3)$$G\left(\vec{r}\right)=\langle B(t)I\left(\vec{r}\right)\rangle -\langle B(t)\rangle \langle I\left(\vec{r}\right)\rangle$$

The target intensity image slice can be expressed as:(4)$$G^{\left(2\right)}\left(\vec{r},t\right)=\langle B(t)I\left(\vec{r},t\right)\rangle -\langle B(t)\rangle \langle I\left(\vec{r},t\right)\rangle$$
where $G^{\left(2\right)}\left(\vec{r},t\right)$ represents the target intensity image slice obtained by a time slice t through the method of ghost imaging, and $I\left(\vec{r},t\right)$ represents the intensity of the speckle light field irradiated on the target at the plane coordinate r and the time slice t. $B\left(t\right)$ represents the sum of the intensity of the light field reflected by the target at the time slice t.

The echo signal of the $t_{n}$ slice received by the single point detector is expressed as:(5)$$B(n)=\sum I\left(\vec{r}\right)T\left(\vec{r}\right)P_{0}\exp\left(-\frac{{\left(t-t_{0}-nT\right)}^{2}}{2a^{2}}\right)$$
where $t_{n}$ represents slice serial numbers. After M measurements, the calculation formula of the target range profile under a different diffraction slice is:(6)$$G_{t_{n}}(x,y)=\frac{1}{M}\sum_{n=1}^{M}\left(B_{n}(I,t_{n})-\langle B\rangle \right)I_{n}(x,y,t_{n})$$

The laser velocity measurement is based on the slice difference. The target moving speed is V and the light speed is c; the detection system will continuously send pulses to the target and receive the echo signal. $R_{1}$ is the distance between the target and the system when the first laser pulse hits the target; the velocity detection formula is based on the difference:(7)$$V=\frac{R_{2}-R_{1}}{\Delta t}$$

We set the target range image at different times as $G_{1}∼G_{N}$; we took the range image at $t_{1}$ time as $G_{1}(x,y)$ a standard value and made a difference between the subsequent range images and $G_{1}$. According to the principle of the slice differential velocity measurement, all the pixels on the multiple range images were differentiated, and the velocity image was calculated after the distance information was obtained.
(8)$$\begin{array}{l} V_{A}(x,y)=\frac{G_{A}(x,y)-G_{1}(x,y)}{\Delta t_{A}}\\ V_{B}(x,y)=\frac{G_{B}(x,y)-G_{1}(x,y)}{\Delta t_{B}}\;\\ \;\;\;\vdots \\ V_{N}(x,y)=\frac{G_{N}(x,y)-G_{1}(x,y)}{\Delta t_{N}} \end{array}$$

Combined with the principle of slice difference and the ghost imaging formula, the detection formula of the correlation imaging velocity profile is provided in this paper:(9)$$G_{image}\left(x,y,z,t\right)=\frac{1}{M}\sum_{N=1}\sum_{i=1j=1}^{M}\Delta I_{1i}\left(x,y,\frac{G_{A}(x,y)-G_{1}(x,y)}{\Delta t_{N}}\right)\Delta I_{2i}\left(z,t_{j}\right)$$

At present, conventional imaging uses an array detector, and the echo light is incident on the array detector through beam splitting. Ghost imaging uses a single point detector to receive the reflected light, and all of them are incident on the detector without beam splitting. Under the same noise condition, the signal-to-noise ratio is greatly improved; that is, the sensitivity is greatly improved. Because it is received by a single point detector, and the echo signal is collected to a point and then received by the detector, the imaging system has the characteristics of a high sensitivity compared with array detection.

## 3. Simulations

According to the above theoretical derivation, the input parameters required for simulation verification are obtained, including setting the target slice interval ΔR, setting the target shape, and setting the target motion state. As shown in Figure 2, the simulation target we selected is a rotating target similar to a fan blade, which rotates clockwise.

Using the pulse formula outlined above that the beam the laser emitted is a Gaussian beam and the random speckle program, the speckle light field hitting the target was saved as a three-dimensional matrix for a subsequent correlation calculation. We calculated the value obtained after each speckle light field acted on the target and returned to the bucket detector. We saved the output value of the barrel detector and prepared to correlate it with the speckle light field to reconstruct the rotating fan target.

Due to the detection of rotating targets, the change in the velocity on the fan was continuous. We obtained the speed of each pixel position on the fan by difference and projected the speed information onto the gray value image of the fan. Different gray information was used to represent the change in the speed of the pixels at different positions. The target rotated clockwise; the speed of the middle part was 0, and the speeds of the left and right sides were positive and negative, respectively.

Figure 3 shows the intensity echo signal value obtained by a bucket detector in one measurement. The light field speckle is correlated with the measured value of the bucket detector to obtain the ghost imaging images at different slices. According to the slice difference method, the ghost imaging detection results with the distance information of the target at different times were obtained, as shown in Figure 4. In the process of a fan blade rotating target detection in this paper, the rotated distance was expressed as the different result of each pixel of different images containing distance information.

The above image resolution is 64 × 64 pixels, and the cumulative calculation times of a single ghost imaging image is 5000 times. The velocity image of the target fan blade can be obtained by the slice differential velocity measurement of the ghost imaging detection results containing distance information at different times. In this paper, different colors represent different speeds of fan blade rotation, and the positive and negative values of the vector velocity represent the velocity direction of different positions of the rotating fan blade. The velocity image results are shown in Figure 5.

## 4. Experiment Results

### 4.1. Experimental System Design

The experimental device of the imaging system is shown in Figure 6 below. The laser source of the experimental device of the imaging system is a pulse laser. The central wavelength of the emitted light is 532 nm, the pulse width is 10 ns, and the repetition frequency is 4 kHz. The mode structure of the laser is TEM00 > 95%, and the root mean square noise is less than 1%. The laser has a stable power and the laser light field has a good uniformity; it can ensure a uniform speckle after beam expansion. It is a laser source with good properties. In order to ensure that the field of view provided by the speckle when the light source hits the object was large enough, a lens with a diameter of 10 mm and a focal length of 20 mm was used at the transmitting end of the experimental system to expand the light emitted by the laser into a speckle field.

The information reflected from the target is received and gathered by the lens. In the experiment, a DMD (digital micromirror device) was selected to modulate the light source (pixel number: 1920 × 1080, pixel size: 13.68 μm). An area of 1024 × 1024 pixels was selected as the modulation module, which was achieved by setting the DMD.

According to the outlined experimental equipment parameters, we know that its repetition frequency is greater than the number of pulses required for laser pulse detection in the object’s motion cycle, and its parameters can ensure multiple measurements within the pulse duration. The DMD that generates speckle has a high conversion frequency, which ensures the detection of objects within the motion cycle of objects. The single point detector used is Pin, which has a high response speed.

Our experimental system used Labview software to accurately control the change in the DMD random speckle and the exposure of a pin single point detector so that the DMD could generate a random speckle in a very short amount of time, which then controls the single-pixel camera to receive a target echo signal. By comprehensively considering the emission speckle control frequency and the response time of the pin detector, we selected the range profile statistics number of 1000 times, and the corresponding maximum velocity measurement value is 0.25 m/s. Our method is not applicable to high-speed targets; it is a new method for the high-precision detection of conventional speed.

As shown in Figure 7, the experimental target selects a “T”-shaped fan blade rotation target, takes the central symmetry axis as the rotation axis, and carries on the rotation operation through the laboratory turntable. According to the theory and simulation part of the content for the laboratory under the condition of “T”-type blades rotating the target correlation imaging speed as in detection experiments, we completed the correlation velocity imaging based on a slice difference experiment.

### 4.2. Analysis of Experimental Results

We obtained the results of the target velocity profile and analyzed the errors in the results. In order to quantitatively evaluate the reconstruction performance of our velocity profile detection method, the root mean square error (*RMSE*) and peak signal-to-noise ratio (PSNR) of the reconstructed image in the obtained velocity profile were calculated. 

Using root mean square distance error (*RMSE*) [14] to evaluate the slice distance error of velocity images at different distances. The *RMSE* is expressed as:(10)$$RMSE=\sqrt{\frac{\sum_{i,j}^{n}{\left(G_{i,j}-O_{i,j}\right)}^{2}}{i\times j}}$$
where $G_{i,j}$ is the range image, $O_{i,j}$ is the real range value of the target. $\left(i,j\right)$ represents the coordinate of range images.

Through the experimental data, we obtained the experimental results of the target correlation imaging with range information at different times and analyzed the accuracy of the range image information at the following two times.

Figure 8 depicts the experimental results of target correlation imaging with distance information at different times. According to the above formula of the root mean square error and peak signal-to-noise ratio, we calculated that the *RMSE* of Figure 8a is $\sigma_{a}=0.015\;m$, the *RMSE* of Figure 8b is $\sigma_{b}=0.021\;m$. The calculated root mean square error of the experiment shows that the range image of the target can correctly reflect the range information of the target. 

Based on the range image obtained above, we performed slicing difference on the range image to obtain the experimental results of associated velocity imaging based on the slicing difference. The velocity image results are shown in Figure 9 below, where the horizontal and vertical coordinates, respectively, represent the pixel resolution of the target, and different depth colors represent the different velocities of each part of the target.

To reflect more experimental data, we used the data to obtain the velocity aberration results at different times, and added the velocity images at different times, as shown in Figure 10 below:

It can be seen from Figure 10 that within the same integration time, the detection system acquired stability for the velocity image of the target. The experimental imaging results correspond to the theoretical and simulation results. What we have outlined here is a new method for the high-precision detection of conventional speed.

## 5. Discussion

From the history of ghost imaging, in 1995, Shi [20] realized ghost imaging through entangled sources for the first time in experiments. In 2002, Bennink [21] achieved ghost imaging using classical light sources. Through our simulation and experimental verification, it has been proven that the classical correlation is sufficient for our detection. Our system mainly focuses on the application of long-range high-performance imaging, therefore, use the classical correlation detection system.

Different from the elimination of motion blur in the ghost imaging of moving targets, this paper proposes a method of velocity image detection via ghost imaging. The system can detect the velocity image of the target directly. Based on the imaging system, a high signal-to-noise ratio and high-resolution imaging can be achieved. However, this excellent imaging method has limitations in terms of the imaging speed due to the need for a statistical calculation. The detection system can be upgraded by using lasers with higher repetition rates and detectors with higher response frequencies. At present, the optimization of its algorithm, such as compression sensing, can greatly improve its imaging speed [22,23,24]. The research on the improvement of its imaging performance is also a research hot spot; more algorithms such as deep learning [25] could be more widely used in this research area. Supported by the above research, the target velocity image detection system based on ghost imaging should be widely used in many future applications.

## 6. Conclusions

In this paper, a new ghost imaging velocity image lidar system based on slice difference is proposed. It overcomes the limitation of the volume and power consumption of traditional lidar for long-distance detection, and it has the characteristics of a high resolution. Through theoretical research, simulation research, and experimental research, the ghost imaging velocity profile of a 64 × 64 pixel target is successfully detected, and its performance is analyzed. Our research shows that this method can reconstruct the velocity profile information of unknown targets, and this research has far-reaching significance for the development of laser active multi-dimensional imaging.

## Figures and Tables

**Figure 1 sensors-23-04255-f001:**
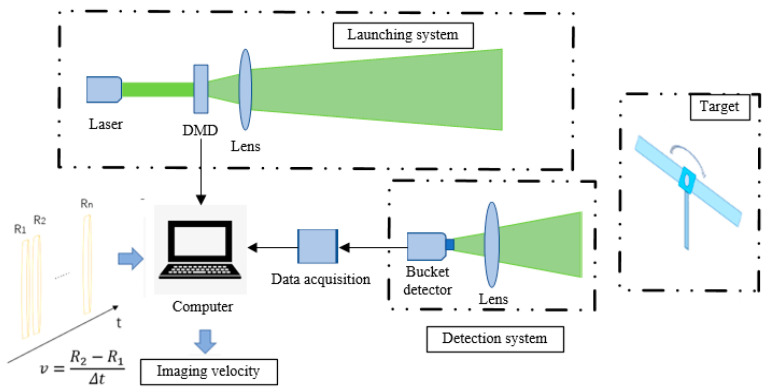
Ghost imaging velocity image detection system based on slice difference.

**Figure 2 sensors-23-04255-f002:**
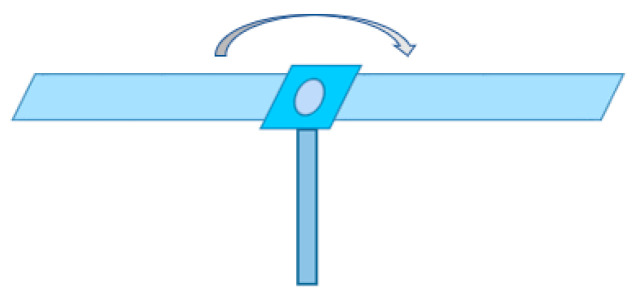
Schematic diagram of rotating target fan.

**Figure 3 sensors-23-04255-f003:**
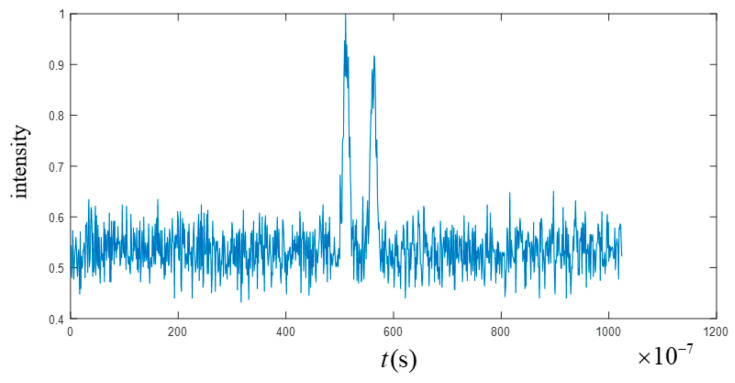
Echo signal value obtained by bucket detector in a certain measurement, where the ordinate is the normalized detection intensity value.

**Figure 4 sensors-23-04255-f004:**
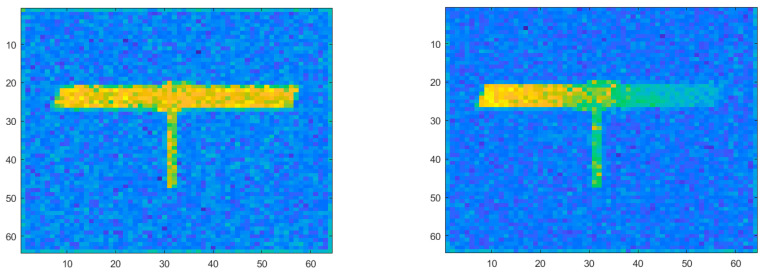
Detection results of ghost imaging with distance information at different times.

**Figure 5 sensors-23-04255-f005:**
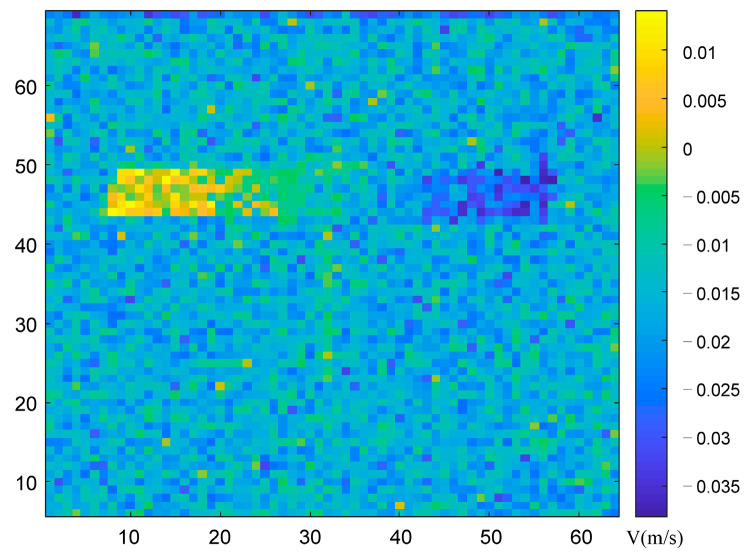
Simulation results of correlation imaging velocity profile detection based on slice difference method.

**Figure 6 sensors-23-04255-f006:**
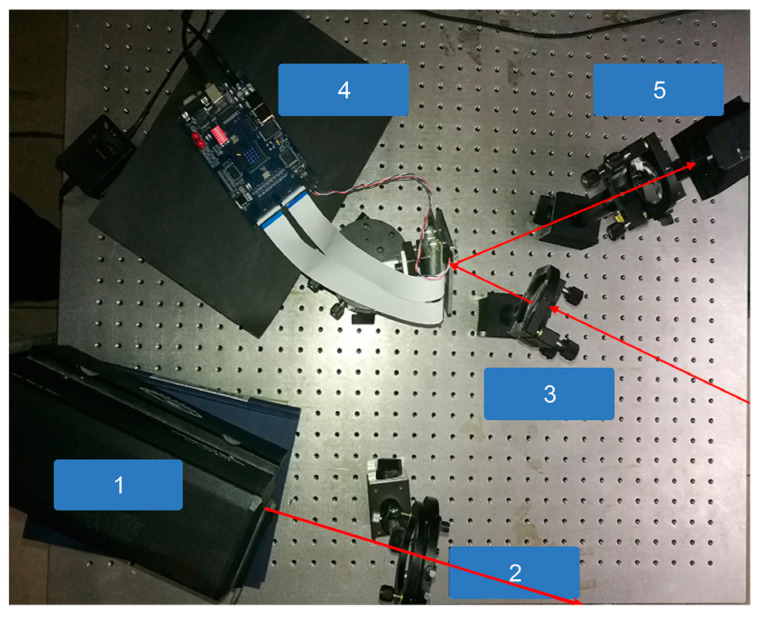
Schematic diagram of experimental launch system: (1) laser (532 nm), (2) beam expander, (3) convergent lens, (4) DMD, (5) PIN single point detector. The red arrow in the figure represents the direction of the light path.

**Figure 7 sensors-23-04255-f007:**
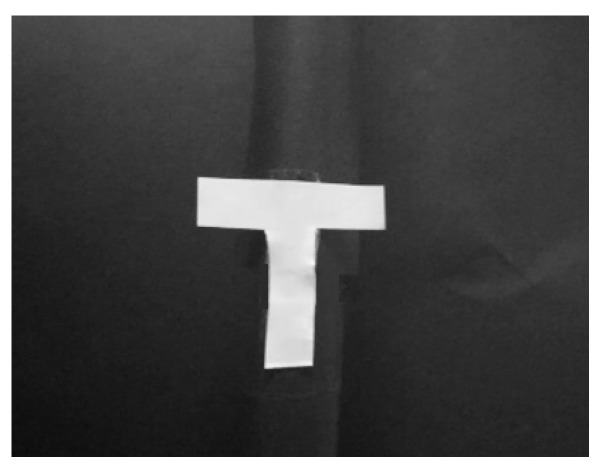
Objectives used in the experiment.

**Figure 8 sensors-23-04255-f008:**
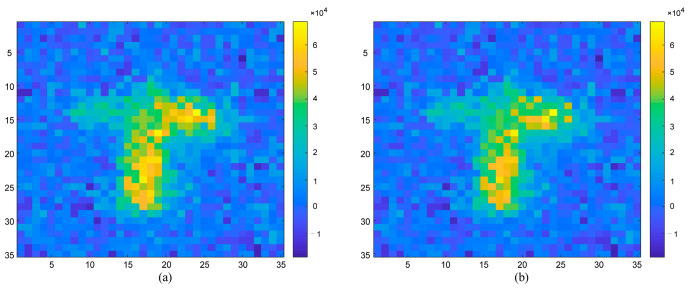
Experimental results of target correlation imaging with distance information at different times. (**a**) Images containing distance information at time *t_a_*. (**b**) Images containing distance information at time *t_b_*.

**Figure 9 sensors-23-04255-f009:**
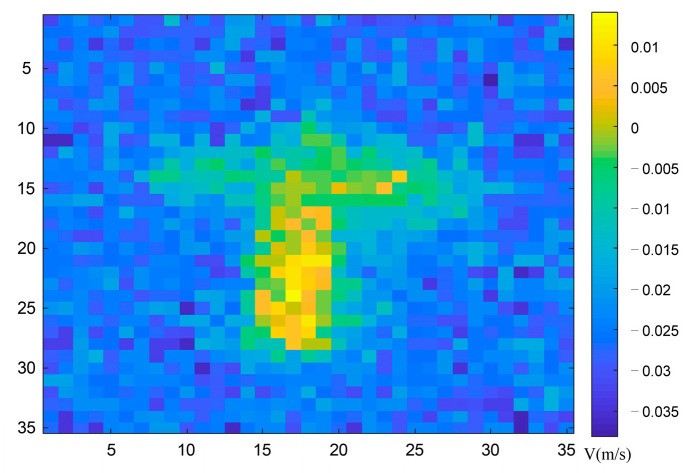
Experimental results of target velocity profile obtained by slice difference.

**Figure 10 sensors-23-04255-f010:**
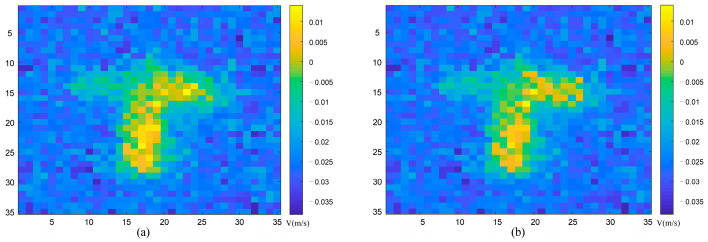
Velocity image results at different times. (**a**) *t_a_* moment velocity profile result; (**b**) *t_b_* moment velocity profile result.

## Data Availability

Data available on request due to privacy.

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
