# Peer review of "Target Velocity Ghost Imaging Using Slice Difference Method"

_sensors, 2023, doi:10.3390/s23094255_

Round 1

Reviewer 1 Report

The article titled “Target velocity ghost imaging via slice difference method” was read carefully and the review report is presented here.

This work, as its title implies is a research article investigating theoretical and experimental imaging system through scattering medium via special algorithm.

The article is in the field of fundamental imaging and measurement science.  In this
aspect it can be considered as  a nice application of laser and optics related research
article.

 I therefore would in principle agree that the manuscript be published
in your Journal , after the many issues raised in this review are all responded
to. Especially the ones concentrating on the experiment itself.

The authors within the framework of their project suggest and implement the detection method of ghost imaging target velocity image based on slice difference. They conjecture that the results show that compared with the array velocity detection system, the system has a simple structure and stronger practical value. They state that they can largely enhance the imaging sensitivity.

Simple typos grammatical mistakes such as the sentence

“At this stage, there is a lack of high-sensitivity velocity image imaging system”

There may be wrong use of vocabulary, please correct.  Proofread manuscript.

Recent articles missing, upto date literature should be cited.

Please cite following reference related work

Moving Target Imaging via Computational Ghost Imaging Combined With Artificial Bee Colony Optimization -IEEE Transactions

Deep learning based methods are finding more ground in optical imaging. Such a coherent imaging deep learning based imaging application was reported recently,

In  “Classification of material type from optical coherence tomography images using deep learning”, International Journal of Optics

The authors could use this reference to also broaden application and increase the sensitivity of their measurements using deep learning algorithms in their imaging setup. Add discussion.

In equation 4 the echo signal of the tn slice received by the single point detector is expressed, is it correct, check and clarify.

It is stated that the light field speckle is correlated with the measured value of the bucket  detector to obtain the ghost imaging images at different slices. According to the slice difference method, the ghost imaging detection results with distance information of the target at different times are obtained, as shown in the Figure 4. In the process of fan blade  rotating target detection in this paper, the rotated distance can be expressed as the difference result of each pixel of different images containing distance information.

What is the imaging resolution of these pixels? Elaborate.

In Figure 3 the echo signal is given.  Does the actual time scale match the distances in the setup?

Clarify with respect to Figure 4 & 5.

Within the experimental realization:

Laser wavelength is stated to 532 nm, and mentioned that it is a laser with good properties, it is not clear, please clarify. The coherence length, and spectral properties.

The pulse width is not given, what is it.

Give the abbreviation DMD.

The calculated  peak signal-to-noise ratio of experiment shows that the range image of the target we get has high image quality.  

SNR image quality was controlled with modehopping and also increased electronically in the articles

Other Points:

Explain clearly the novelty of the work. This should be given in the abstract and main body. Not just as an extension to previous study but underline originality of this present manuscript..

Some Figure  is not properly visible. Prepare a clearer depiction if possible,

increase resolution.  (if possible for example fig 7)

What are the limitations of the study? Elaborate.

Clarify theoretical-experimental bridge.

Compare  with state of the art or other relevant methods?

Add relevant mentioned references above and other..

Give recent references (from 2021-2022) especially for state of art.

Please elaborate the outlook and comment more on real life effects on practical imaging.

I  would therefore suggest its recommendation to be published in your  journal, once all
the issues raised in this review report are responded to.

Author Response

Dear reviewer:

On behalf of our coauthors, we would like to thank you again for considering our manuscript and for your careful handling of our submission. As requested, we have addressed each and every single comment and question raised by the referees in a point-by-point fashion below. In addition, we have highlighted all the associated changes made to the manuscript. 

We are thankful to the referees for their detailed reports. Indeed, the referees provided comments that enabled us to provide stronger motivations and to include additional calculations in the new version of our manuscript. We are certain that the current manuscript will be well-received by readers interested in the field of ghost imaging.

We believe that we have successfully addressed each comment from referees and consequently, we feel that our manuscript is now suitable for publication in Sensors.

Reviewer 2 Report

In this paper, the ghost imaging technology based on the slice differential method was used to measure the velocity of moving targets, which provided a feasible method for detecting multi-dimensional information.  However, it still lacked some very critical components and demonstrations to be acceptable in Sensors journal. I would like to see the authors’ responses and revisions according to the following comments before a decision can be made.  

 1.       Ghost imaging technology requires multiple measurements to reconstruct 3D or more information of the static target with higher accuracy, that is, the more measured times, the better quality of the object information recovery can be retrieved. For moving objects, recovering the target’s information of an object at a certain instant means detecting dynamic scene with a  short-time laser pulse which is shorter than the object’s moving period, that is, multiple measurements in a pulse duration, the position information of the object in the first pulse durationcan be recovered. Sequentially, the positional information of the object in the second pulse can be reconstructed by multiple measurements in the second pulse. Therefore, it is necessary to accurately control the synchronization of the pulse time with the exposure time of the CCD and the exposure time of the single-pixel camera. However, the introduction of synchronous control with the laser emitting and CCD are not given in the paper, that is, the experimental details such as the parameters of laser pulse, exposure time of CCD and etc.  are not clearly described. In addition, the target located information obtained by this method is acquired by the average positions within a pulse duration, thus the measured accuracy is not convicing.

2.       Please introduce the slice difference method in detailed.

3.       In line 36: In recent years, there have been several articles about the method of ghost imaging velocity measurement. It is recommended to cite more literatures.

4.       In line 44: It is said in the article that 'imaging sensitivity can be greatly improved', but the full text lacks in introducing the specific sensitivity and the calculation formula of sensitivity is not given.

5.       Lines 52-55:It is suggested to draw a schematic diagram for clearly explanation.

6.       Lines 62 and 66: The formula and text are not in the same line. Please check the whole article

7.       Line 121: No unit after '5000'.

8.       Line 139: There is no 'T' fan as the detection object and CCD and other devices in the figure. 6. Please update the experimental diagram.

9.       Lines 166 and 167: Figures (a) and Figure (b) cannot be found in the full text.

10.    Lines 168-171: The calculation of RMSE and PSNR requires the reference picture, but the reference picture is not given in the text, and the reference picture is not described accordingly.

11.    Line 178: The coordinate axis of Figure (7) is fuzzy.

12.    Line 182: 'The system has the advantage of super-resolution, the resolution of other methods is not mentioned in the article, and it is impossible to compare whether it has better resolution. Moreover, from the effect of Figure (7), the imaging quality is not good.

13.    There is little experimental data, so speed images at different times should be supplemented for comparison.

Author Response

(The authors gave the same response as above.)

Reviewer 3 Report

This paper proposes a method of velocity image detection using ghost imaging. The authors present compelling evidence of the performance of this system, with a high resolution and the sensible advantage of  overcoming the limitation of volume and power consumption of traditional lidar for long-distance detection. Therefore, I think the results deserve publication.

However, I have a few concerns about the contents of the paper. I suggest the authors to consider these points to improve the quality of the paper.

- The question of ghost imaging has a long history and I still remember heated debates about the classical or quantum nature of this technique. Especially, when bucket detectors are involved. Is it enough a classical correlation between the beams or do we need quantum one? I suggest the authors to elaborate about this topic.

- Since ghost imaging was initially tailored at the single-photon level, can this technique be used with singel-photon deterministic sources?

- I see a bottleneck in the proposed method: the speed due to the requirement of statistical analysis. Can the authors comment on possible solutions to this serious drawback?

Once these points are properly addressed, I will have no hesitation in recommending publication.  

Author Response

(The authors gave the same response as above.)

Round 2

Reviewer 2 Report

The authors have addressed all my issues, I can recommend the revised manuscript for publication.

Author Response

Thank you for your approval of the content of my manuscript and the revised response. Also thanks for the recommendation.

Reviewer 3 Report

The authors have carefully addressed my concerns. I have no hesitation in recommending publication.